# Adaptive Prescribed Performance Control of Robotic Manipulators with Velocity Constraints and Arbitrary Initial Joint Positions

Xing Ren
*School of Automation Engineering*
*University of Electronic Science and Technology of China*
Chengdu 611731, China
renxing@std.uestc.edu.cn

Qing Guo
*School of Aeronautics and Astronautics*
*University of Electronic Science and Technology of China*
Chengdu 611731, China
guoqinguestc@uestc.edu.cn

Tieshan Li
*School of Automation Engineering*
*University of Electronic Science and Technology of China*
Chengdu 611731, China
tieshanli@126.com

Xinyu Li
*College of Electronics Information Engineering*
*Hebei University*
Baoding 071002, China
ce_inu6@126.com

*Abstract*—A novel adaptive prescribed performance control method for the $n$-DOF robotic manipulators is proposed in this paper. Firstly, a normalization constraining function $g$ is designed for state transformation and constructing a new system model. The risk of singularity problems can be reduced by adjusting the parameters of function $g$. Then, an adaptive prescribed performance controller is developed based on the constructed system model, which allows the initial joint positions of the manipulator to be arbitrary, and can guarantee that the tracking error reaches the desired accuracy within the preset time $T_P$. Meanwhile, different velocity constraints can be individually imposed on each joint. In addition, an adaptive law is designed to estimate the lumped uncertainty in the system and compensate for its negative influence, which improves the control performance. Finally, the effectiveness of the proposed method is verified by simulations and experiments on a 4-DOF manipulator.

*Index Terms*—Robotic manipulator, prescribed performance control, state constraints, arbitrary initial positions, adaptive estimation law.

## I. INTRODUCTION

Robotic manipulators, as essential components of automation and robotics technology, have demonstrated their enormous potential and extensive application value in many fields [1]–[5]. Due to the significant increase in complexity and difficulty of control tasks, the performance requirements for robotic manipulators have correspondingly increased, especially for control accuracy. Many researchers have studied the high-accuracy control of robotic manipulators, such as Liu *et al.* [6] proposed two adaptive bias radial basis function neural network (RBFNN) control schemes: local bias scheme and global bias scheme, which can improve the approximation accuracy of the RBFNN and the control performance. Zhong

This work was supported by the National Natural Science Foundation of China under Grant 52175046 and Grant 51939001. (Corresponding author: Qing Guo.)

*et al.* [7] designed a control scheme that combines PID control and fast terminal sliding mode control (TSMC) for a redundant manipulator, which improves both the control performance and the system robustness. Meng *et al.* [8] presented an adaptive finite-time command filtered backstepping control method for manipulators with unknown backlash, the explosion of complexity is avoided, and the tracking error can converge to a neighborhood of the origin within finite time. Rahmani *et al.* [9] proposed a novel sliding mode control method with a new sliding surface, and an extended grey wolf optimizer is used to tune the controller parameters. In addition, a backstepping sliding mode controller with a new integral sliding manifold and a new nonlinear disturbance observer is developed by Xi *et al.* [10] for robot manipulators.

However, the methods mentioned above can only guarantee that the tracking error ultimately converges to a neighborhood of the origin, and the tracking performance cannot be accurately quantified or prescribed. Thus many studies have focused on the prescribed performance control for the manipulators. For example, Song *et al.* [11] investigated an adaptive prescribed performance control scheme for manipulators with input constraint and external disturbances, where a new accumulated error vector integrated with a performance enhancement function is constructed to achieve the preset accuracy. Sai *et al.* [12] proposed an approximate continuous fixed-time TSMC scheme based on the designed prescribed performance function (PPF), which can improve the transient and steady-state performance of the trajectory tracking. Lyu *et al.* [13] used two auxiliary functions to transform the tracking error, and embedded it into the barrier Lyapunov function to achieve the prescribed performance control. Besides, Sun *et al.* [14] proposed a prescribed performance control strategy based on the fixed-time non-singular TSMC and the PPF, and built

an auxiliary system to compensate for the negative influence of the input saturation.

Although these methods can achieve prescribed performance control of the robotic manipulators, the initial tracking errors of the joints are required to satisfy certain conditions, which is very unfavorable for actual systems with unknown initial states. Therefore, one motivation of this paper is to solve this problem. Furthermore, the manipulators are often affected by model uncertainties and unknown friction, which also brings about a significant challenge to achieving high-precision control [15], [16]. Inspired by the above studies and existing problems, a novel adaptive prescribed performance control method for the $n$-DOF robotic manipulators is proposed in this paper, and the main contributions are as follows:

1) A normalization constraining function $g$ is designed, which can represent the distance between the input variable and its boundary. Then a new manipulator model is constructed using the function $g$, and changing the parameters of $g$ can reduce the risk of singularity.

2) Based on the constructed model, a prescribed performance control method is proposed, which is independent of the initial joint positions of the manipulator, and can guarantee that the tracking error enters the desired performance within the preset time $T_p$.

3) The constraint of joint velocity can be consistently satisfied by using the developed controller. Besides, an adaptive law is designed to estimate the lumped uncertainty in the system and compensate for its negative impact. The effectiveness of the proposed method is verified through both simulations and experiments.

## II. PROBLEM FORMULATION AND PRELIMINARIES

### A. Model of Robotic Manipulator

The dynamic model of a rigid manipulator with $n$-DOF can be described as

$$
\begin{aligned}
\ddot{\mathbf{q}} =& \mathbf{M}^{-1}\left(\mathbf{q}\right)\boldsymbol{\tau} + \mathbf{M}^{-1}\left(\mathbf{q}\right)\left(-\mathbf{C}\left(\mathbf{q},\dot{\mathbf{q}}\right)\dot{\mathbf{q}} - \mathbf{G}\left(\mathbf{q}\right)\right) \\
&+ \mathbf{M}^{-1}\left(\mathbf{q}\right)\left(-\mathbf{F}\left(\dot{\mathbf{q}}\right) - \boldsymbol{\Delta}\right)
\end{aligned}
\tag{1}
$$

where $\mathbf{q}, \dot{\mathbf{q}}, \ddot{\mathbf{q}} \in \mathbb{R}^n$ denote the position, velocity, and acceleration of the joints. $\mathbf{M}\left(\mathbf{q}\right) \in \mathbb{R}^{n \times n}$ represents the inertia matrix, which is symmetric and positive definite, $\mathbf{C}\left(\mathbf{q},\dot{\mathbf{q}}\right)$ is the Centripetal and Coriolis matrix, $\mathbf{G}\left(\mathbf{q}\right)$ is the gravitational torque vector, $\mathbf{F}\left(\dot{\mathbf{q}}\right)$ is the friction, $\boldsymbol{\Delta}$ is the model uncertainty, and $\boldsymbol{\tau}$ is the driving torque. In this paper, the arguments in functions or matrices sometimes are sometimes omitted if no confusion can arise from the context.

If the state vector is defined as $\left[\mathbf{x}_1, \mathbf{x}_2\right]^T = \left[\mathbf{q}, \dot{\mathbf{q}}\right]^T$, then the robotic manipulator model can be written as

$$
\begin{cases}
\dot{\mathbf{x}}_1 = \mathbf{x}_2 \\
\dot{\mathbf{x}}_2 = \mathbf{M}^{-1}\boldsymbol{\tau} + \mathbf{f}\left(\mathbf{x}_1, \mathbf{x}_2\right) + \mathbf{d}\left(\mathbf{x}_1, \mathbf{x}_2\right)
\end{cases}
\tag{2}
$$

where $\mathbf{f} = \mathbf{M}^{-1}\left(-\mathbf{C}\mathbf{x}_2 - \mathbf{G}\right)$ is a known function, $\mathbf{d} = \mathbf{M}^{-1}\left(-\mathbf{F} - \boldsymbol{\Delta}\right)$ is the unknown lumped uncertainty, and

$\mathbf{x}_j = \left[x_{j,1}, \cdots, x_{j,n}\right]^T$ $(j = 1, 2)$. The performance function $\lambda_i(t)$ for the $i$-th joint of the manipulator is considered as

$$
\lambda_i(t) = \left(\lambda_{i,0} - \lambda_{i,\infty}\right) \cdot \exp\left(-\gamma_i t\right) + \lambda_{i,\infty}
\tag{3}
$$

where $\lambda_{i,0}$ is the initial performance, $\lambda_{i,\infty}$ is the ultimate performance $(\lambda_{i,0} \geq \lambda_{i,\infty} > 0)$, and $\gamma_i$ is a positive constant.

The main task of this paper is to achieve the prescribed performance control of the n-DOF manipulators with velocity constraints, and the initial positions of all joints are allowed to be arbitrary, i.e., for any given performance function $\lambda_i(t)$ and preset time $T_p$, the tracking error satisfies $|e_i(t)| = |x_{d,i}(t) - x_{1,i}(t)| < \lambda_i(t)$, $t \geq T_p$, where $x_{d,i}(t)$ is the desired angular trajectory of the $i$-th joint. Different from other works, $|e_i(0)| < \lambda_{i,0}$ is not necessary for our method. Meanwhile, the velocity $x_{2,i}$ satisfies the following constraint:

$$
-c_{2,i} < x_{2,i}(t) < c_{2,i}, \ t \geq 0
\tag{4}
$$

where $c_{2,i}$ is the constraint boundary designed by users, and $c_{2,i}$ can be set to $\infty$ to remove the corresponding constraint.

*Remark 1:* The vector $\boldsymbol{\Delta}$ can be expressed as $\boldsymbol{\Delta} = \Delta\mathbf{M}\ddot{\mathbf{q}} + \Delta\mathbf{C}\dot{\mathbf{q}} + \Delta\mathbf{G}$, where $\Delta\mathbf{M}$, $\Delta\mathbf{C}$, and $\Delta\mathbf{G}$ are the uncertain parts of the model. In addition, the lumped uncertainty $\mathbf{d}$ and its first-order derivative $\dot{\mathbf{d}}$ are bounded such that $\|\mathbf{d}\| \leq \bar{d}_1$ and $\left\|\dot{\mathbf{d}}\right\| \leq \bar{d}_2$, where $\bar{d}_1$ and $\bar{d}_2$ are unknown positive constants.

*Assumption 1:* The desired trajectory $x_{d,i}(t)$ is continuous and bounded, and its first-order and second-order derivatives are also bounded, such that $|x_{d,i}(t)| \leq \bar{d}_{1,i}$, $|\dot{x}_{d,i}(t)| \leq \bar{d}_{2,i}$, and $|\ddot{x}_{d,i}(t)| \leq \bar{d}_{3,i}$, where $\bar{d}_{1,i}$, $\bar{d}_{2,i}$, and $\bar{d}_{3,i}$ are unknown positive constants.

### B. Preliminaries

*Lemma 1 ( [17]):* Consider the following system:

$$
\dot{x} = f(t, x), \qquad x(0) = x_0
\tag{5}
$$

where $x \in \mathbb{R}$ is the system state, and $f(\cdot)$ is a continuous nonlinear function. There exists a continuous funciton $V(x(t)) \geq 0$ defined on $t \in [0, \infty)$ with bounded initial value $V(0)$. If the following inequality holds:

$$
\dot{V}(t) \leq -aV(t) + b
\tag{6}
$$

where $a$ and $b$ are two positive constants, then the solutions of system (5) are uniformly ultimately bounded (UUB), and $V(t) \leq b/a$ as $t \to \infty$.

*Lemma 2 ( [18]):* For any given $\chi_1 \geq 0$ and $\chi_2 \geq 0$, if $p > 1$ and $q > 1$ are real numbers satisfying $1/p + 1/q = 1$, then the following inequality (i.e., Young's inequality) holds:

$$
\chi_1\chi_2 \leq c\frac{\chi_1^p}{p} + c^{-\frac{q}{p}}\frac{\chi_2^q}{q}
\tag{7}
$$

where $c$ is a positive scalar.

## III. MAIN RESULTS

### A. System Model Transformation

Firstly, a normalization constraining function $g(x, a, b)$ is designed as follows to represent the distance between the input variable and its boundary:

$$g(x,a,b) = \begin{cases} 0, & x \leq -ab \\ \left(1 - \left(\dfrac{x}{ab}\right)^2\right)^3 \cdot \exp\left[0.5\left(\dfrac{x}{ab}\right)^5\right], \\ & -ab < x < 0 \\ \left(1 - \left(\dfrac{x}{ab}\right)^2\right)^3 \cdot \exp\left[-0.5\left(\dfrac{x}{ab}\right)^5\right], \\ & 0 \leq x < ab \\ 0, & x \geq ab \end{cases} \tag{8}$$

where $x$ is the input variable, $b$ is the boundary, and $a \geq 1$ is a parameter that can adjust the shape of the function $g$. Both $a$ and $b$ can be time-varying or constant. The partial derivative of function $g$ with respect to $x/ab$ is

$$g_p(x,a,b) = \frac{\partial g}{\partial\left(\frac{x}{ab}\right)}$$

$$= \begin{cases} 0, & x \leq -ab \\ -6\left(\dfrac{x}{ab}\right)\left(1 - \left(\dfrac{x}{ab}\right)^2\right)^2 \cdot \exp\left[0.5\left(\dfrac{x}{ab}\right)^5\right] \\ \quad +2.5\left(1 - \left(\dfrac{x}{ab}\right)^2\right)^3\left(\dfrac{x}{ab}\right)^4 \cdot \exp\left[0.5\left(\dfrac{x}{ab}\right)^5\right], \\ & -ab < x < 0 \\ -6\left(\dfrac{x}{ab}\right)\left(1 - \left(\dfrac{x}{ab}\right)^2\right)^2 \cdot \exp\left[-0.5\left(\dfrac{x}{ab}\right)^5\right] \\ \quad -2.5\left(1 - \left(\dfrac{x}{ab}\right)^2\right)^3\left(\dfrac{x}{ab}\right)^4 \cdot \exp\left[-0.5\left(\dfrac{x}{ab}\right)^5\right], \\ & 0 \leq x < ab \\ 0, & x \geq ab \end{cases} \tag{9}$$

Then two new state variables $\psi_{1,i}$ and $\psi_{2,i}$ are defined as follows [19]:

$$\psi_{1,i} = \frac{e_i}{g_{1,i}}, \quad \psi_{2,i} = \frac{x_{2,i}}{g_{2,i}} \tag{10}$$

with

$$g_{1,i} = g(e_i, h_{1,i}, \lambda_i), \quad g_{2,i} = g(x_{2,i}, h_{2,i}, c_{2,i}) \tag{11}$$

where $h_{1,i}$ and $h_{2,i}$ are design parameters. The curve of $g_{1,i}$ with different $h_{1,i}$ is shown in Fig. 1 (a). We can see that $g_{1,i}$ is continuous and bounded. When $e_i(t) = 0$, then $g_{1,i} = 1$, and the state variable $\psi_{1,i} = e_i(t)$. If $h_{1,i} = 1$, then $g_{1,i} = 0$ at $e_i(t)/\lambda_i(t) = 1$ ($e_i(t)$ on the boundary), while if $h_{1,i} > 1$, then $g_{1,i} > 0$ at $e_i(t)/\lambda_i(t) = 1$ and is proportional to $h_{1,i}$. The curve of $g_{2,i}$ is omitted here due to similarity. It is worth noting that $h_{1,i}$ plays a key role in the control performance of the system. In this paper, $h_{1,i}$ is designed in the following time-varying form:

$$h_{1,i} = (h_{u,i} - 1)\,\mu(t) + 1 \tag{12}$$

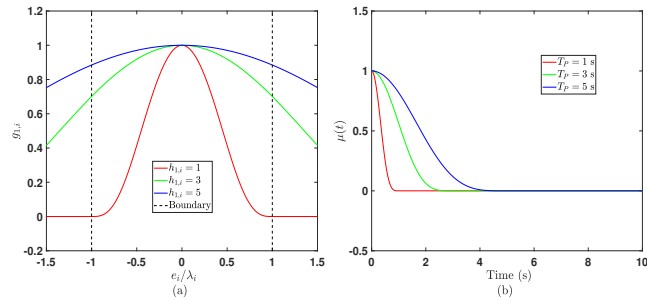

Fig. 1. (a) Curve of $g_{1,i}$ relative to $e_i(t)/\lambda_i(t)$ with different $h_{1,i}$. (b) Curve of $\mu(t)$ relative to time $t$ with different $T_p$.

where $h_{u,i} > 1$ is a sufficiently large constant, and $\mu(t)$ is a time adjustment function, which is constructed as

$$\mu(t) = \begin{cases} 0.25\left(1 + \cos\left(\dfrac{\pi t}{T_p}\right)\right)^2, & 0 \leq t \leq T_p \\ 0, & t > T_p \end{cases} \tag{13}$$

where $T_p$ is a user-defined time constant. The curve of function $\mu(t)$ with different $T_p$ is shown in Fig. 1 (b). Obviously, $\mu(t)$ is bounded, continuous, and monotonically decreases from 1 to 0 within $t \in [0, T_p]$. Thus $h_{1,i}$ is monotonically decreasing for $t \in [0, T_p]$ with $h_{1,i}(0) = h_{u,i}$ and $h_{1,i}(T_p) = 1$, and remains unchanged at 1 after $t > T_p$. Now we can obtain the following Lemma.

*Lemma 3:* For any initial joint position $x_{1,i}(0)$, if $\psi_{1,i}$ satisfies $\psi_{1,i} \in L_\infty$, then $-\lambda_i(t) < e_i(t) < \lambda_i(t)$ holds for $t \in [T_p, +\infty)$. For initial velocity satisfying $-c_{2,i} < x_{2,i}(0) < c_{2,i}$, and $h_{2,i}$ is designed as $h_{2,i} = 1$, if $\psi_{2,i} \in L_\infty$, then $-c_{2,i} < x_{2,i}(t) < c_{2,i}$ holds for $t \in [0, +\infty)$.

*Proof:* From (12), we have $h_{1,i} = 1$, $t \geq T_p$, so $g_{1,i} = 0$ for $|e_i(t)| = \lambda_i(t)$, $t \geq T_p$. Assume there exists a time $t_1 \geq T_p$ such that $e_i(t_1) \geq \lambda_i(t_1)$ or $e_i(t_1) \leq -\lambda_i(t_1)$. Since $e_i(t)$ is continuous, based on the intermediate value theorem, there exists a time $T_p \leq t_2 \leq t_1$ such that $e_i(t_2) = \lambda_i(t_2)$ or $e_i(t_2) = -\lambda_i(t_2)$, thus $g_{1,i}(e_i(t_2), h_{1,i}(t_2), \lambda_i(t_2)) = 0$ and $\psi_{1,i} = \infty$, which is contradictory to $\psi_{1,i} \in L_\infty$. Therefore, $-\lambda_i(t) < e_i(t) < \lambda_i(t)$ for $t \in [T_p, +\infty)$. The proofs for $x_{2,i}(t)$ is similar and will not be repeated here. ∎

*Remark 2:* If $h_{u,i}\lambda_{i,0}$ is greater than the maximum physical limit of the tracking error $e_i$, then $h_{u,i}$ is large enough to allow the initial position $x_{1,i}(0)$ to be arbitrary.

From *Lemma 3*, guaranteeing that $\psi_{1,i}$ and $\psi_{2,i}(i = 1, \cdots, n)$ are bounded is sufficient to achieve the prescribed performance control and velocity constraints mentioned above. Now build a new general model of the manipulator with $\psi_{1,i}$ and $\psi_{2,i}$ as states. Taking the derivative of $\psi_{1,i}$ and $\psi_{2,i}$ yields

$$\begin{cases} \dot{\psi}_{1,i} = -\omega_{1,i}g_{2,i}\psi_{2,i} + \beta_{1,i} \\ \dot{\psi}_{2,i} = \omega_{2,i}\dot{x}_{2,i} + \beta_{2,i} \end{cases} \tag{14}$$

where

$$\omega_{1,i} = \frac{1}{g_{1,i}} - \frac{\partial g_{1,i}}{\partial \left(\frac{e_i}{h_{1,i}\lambda_i}\right)} \frac{e_i}{h_{1,i}\lambda_i g_{1,i}^2}$$

$$= \frac{1}{g_{1,i}} - g_{p1,i}\frac{e_i}{h_{1,i}\lambda_i g_{1,i}^2}$$

$$\beta_{1,i} = \frac{\partial g_{1,i}}{\partial \left(\frac{e_i}{h_{1,i}\lambda_i}\right)} \frac{\dot{h}_{1,i}e_i^2}{\lambda_i h_{1,i}^2 g_{1,i}^2} + \frac{\partial g_{1,i}}{\partial \left(\frac{e_i}{h_{1,i}\lambda_i}\right)} \frac{\dot{\lambda}_i e_i^2}{h_{1,i}\lambda_i^2 g_{1,i}^2}$$

$$+ \left(\frac{1}{g_{1,i}} - \frac{\partial g_{1,i}}{\partial \left(\frac{e_i}{h_{1,i}\lambda_i}\right)} \frac{e_i}{h_{1,i}\lambda_i g_{1,i}^2}\right)\dot{x}_{d,i}$$

$$= g_{p1,i}\frac{\dot{h}_{1,i}e_i^2}{\lambda_i h_{1,i}^2 g_{1,i}^2} + g_{p1,i}\frac{\dot{\lambda}_i e_i^2}{h_{1,i}\lambda_i^2 g_{1,i}^2}$$

$$+ \left(\frac{1}{g_{1,i}} - g_{p1,i}\frac{e_i}{h_{1,i}\lambda_i g_{1,i}^2}\right)\dot{x}_{d,i}$$

$$\omega_{2,i} = \frac{1}{g_{2,i}} - \frac{\partial g_{2,i}}{\partial \left(\frac{x_{2,i}}{h_{2,i}c_{2,i}}\right)} \frac{x_{2,i}}{h_{2,i}c_{2,i}g_{2,i}^2}$$

$$= \frac{1}{g_{2,i}} - g_{p2,i}\frac{x_{2,i}}{h_{2,i}c_{2,i}g_{2,i}^2}$$

$$\beta_{2,i} = \frac{\partial g_{2,i}}{\partial \left(\frac{x_{2,i}}{h_{2,i}c_{2,i}}\right)} \frac{\dot{h}_{2,i}x_{2,i}^2}{c_{2,i}h_{2,i}^2 g_{2,i}^2} + \frac{\partial g_{2,i}}{\partial \left(\frac{x_{2,i}}{h_{2,i}c_{2,i}}\right)} \frac{\dot{c}_{2,i}x_{2,i}^2}{h_{2,i}c_{2,i}^2 g_{2,i}^2}$$

$$= g_{p2,i}\frac{\dot{h}_{2,i}x_{2,i}^2}{c_{2,i}h_{2,i}^2 g_{2,i}^2} + g_{p2,i}\frac{\dot{c}_{2,i}x_{2,i}^2}{h_{2,i}c_{2,i}^2 g_{2,i}^2} \tag{15}$$

with

$$g_{p1,i} = g_p(e_i, h_{1,i}, \lambda_i), \ g_{p2,i} = g_p(x_{2,i}, h_{2,i}, c_{2,i}). \tag{16}$$

Then combined with (2), we can obtain the new model

$$\begin{cases} \dot{\boldsymbol{\psi}}_1 = -\boldsymbol{\Omega}_1\mathbf{G}_2\boldsymbol{\psi}_2 + \boldsymbol{\beta}_1 \\ \dot{\boldsymbol{\psi}}_2 = \boldsymbol{\Omega}_2\left(\mathbf{M}^{-1}\boldsymbol{\tau} + \mathbf{f} + \mathbf{d}\right) + \boldsymbol{\beta}_2 \end{cases} \tag{17}$$

where $\boldsymbol{\psi}_j = [\psi_{j,1}, \cdots, \psi_{j,n}]^T$, $\boldsymbol{\Omega}_j = \text{diag}\{\omega_{j,i}\}$, $\boldsymbol{\beta}_j = [\beta_{j,1}, \cdots, \beta_{j,n}]^T$, and $\mathbf{G}_2 = \text{diag}\{g_{2,i}\}$ $(j = 1, 2; \ i = 1, \cdots, n)$.

### B. Adaptive Prescribed Performance Controller Design

Based on the model (17), we now develop an adaptive prescribed performance controller for $n$-DOF manipulators. Define the error variables as follows:

$$\begin{aligned} \mathbf{z}_1 &= \boldsymbol{\psi}_1, \\ \mathbf{z}_2 &= \boldsymbol{\psi}_2 - \boldsymbol{\alpha}_1 \end{aligned} \tag{18}$$

where $\boldsymbol{\alpha}_1$ is a virtual control law that will be designed later.

*Step 1:* The derivative of $\mathbf{z}_1$ can be written as

$$\begin{aligned} \dot{\mathbf{z}}_1 &= \dot{\boldsymbol{\psi}}_1 \\ &= -\boldsymbol{\Omega}_1\mathbf{G}_2\boldsymbol{\psi}_2 + \boldsymbol{\beta}_1 \\ &= -\boldsymbol{\Omega}_1\mathbf{G}_2\left(\mathbf{z}_2 + \boldsymbol{\alpha}_1\right) + \boldsymbol{\beta}_1. \end{aligned} \tag{19}$$

The virtual control law $\boldsymbol{\alpha}_1$ is designed as

$$\boldsymbol{\alpha}_1 = \left(\boldsymbol{\Omega}_1\mathbf{G}_2\right)^{-1}\left(\mathbf{K}_1\mathbf{z}_1 + \boldsymbol{\beta}_1\right) \tag{20}$$

where $\mathbf{K}_1 = \text{diag}\{k_{1,i}\}$ $(i = 1, \cdots, n)$ is a gain matrix. Consider the following Lyapunov function:

$$V_1 = \frac{1}{2}\mathbf{z}_1^T\mathbf{z}_1. \tag{21}$$

Taking the derivative of $V_1$ and substituting $\boldsymbol{\alpha}_1$ into it yields

$$\begin{aligned} \dot{V}_1 &= \mathbf{z}_1^T\dot{\mathbf{z}}_1 \\ &= -\mathbf{z}_1^T\boldsymbol{\Omega}_1\mathbf{G}_2\left(\mathbf{z}_2 + \boldsymbol{\alpha}_1\right) + \mathbf{z}_1^T\boldsymbol{\beta}_1 \\ &= -\mathbf{z}_1^T\mathbf{K}_1\mathbf{z}_1 - \mathbf{z}_1^T\boldsymbol{\Omega}_1\mathbf{G}_2\mathbf{z}_2. \end{aligned} \tag{22}$$

*Step 2:* The derivative of $\mathbf{z}_2$ is

$$\begin{aligned} \dot{\mathbf{z}}_2 &= \dot{\boldsymbol{\psi}}_2 - \dot{\boldsymbol{\alpha}}_1 \\ &= \boldsymbol{\Omega}_2\left(\mathbf{M}^{-1}\boldsymbol{\tau} + \mathbf{f} + \mathbf{d}\right) + \boldsymbol{\beta}_2 - \dot{\boldsymbol{\alpha}}_1. \end{aligned} \tag{23}$$

The lumped uncertainty $\mathbf{d}$ is estimated by an adaptive law in this paper, which is designed as

$$\dot{\hat{\mathbf{d}}} = \mathbf{K}_d\boldsymbol{\Omega}_2\mathbf{z}_2 - \mathbf{B}_d\hat{\mathbf{d}} \tag{24}$$

where $\mathbf{K}_d = \text{diag}\{k_{d,i}\}$ and $\mathbf{B}_d = \text{diag}\{b_{d,i}\}(i = 1, \cdots, n)$ are two parameter matrices, and $\mathbf{B}_d - \mathbf{I}_n$ is positive definite, where $\mathbf{I}_n$ is the identity matrix. The estimation error is defined as $\tilde{\mathbf{d}} = \mathbf{d} - \hat{\mathbf{d}}$. Then the adaptive prescribed performance controller is designed as

$$\begin{aligned} \boldsymbol{\tau} &= -\mathbf{M}\left(\mathbf{f} + \hat{\mathbf{d}}\right) + \mathbf{M}\boldsymbol{\Omega}_2^{-1}\left(\dot{\boldsymbol{\alpha}}_1 - \boldsymbol{\beta}_2\right) - \mathbf{M}\boldsymbol{\Omega}_2^{-1}\mathbf{K}_2\mathbf{z}_2 \\ &\quad + \mathbf{M}\boldsymbol{\Omega}_2^{-1}\mathbf{G}_2\boldsymbol{\Omega}_1\mathbf{z}_1. \end{aligned} \tag{25}$$

where $\mathbf{K}_2 = \text{diag}\{k_{2,i}\}$ $(i = 1, \cdots, n)$ is a gain matrix.

*Theorem 1:* For the robotic manipulator (2) with any initial joint positions, if $|x_{2,i}(0)| < c_{2,i}$, $h_{1,i}$ is designed as (12), and $h_{2,i} = 1(i = 1, \cdots, n)$. Then the developed controller (25) with adaptive law (24) can guarantee that: 1) The error variables $\mathbf{z}_1$ and $\mathbf{z}_2$, and the estimation error $\tilde{\mathbf{d}}$ are UUB; 2) With the designed performance function $\lambda_i(t)$ and the preset physically feasible time $T_p$, the prescribed performance control, i.e., $|e_i(t)| < \lambda_i(t)$, $t \geq T_p$, can be achieved; 3) The constraint on velocity, i.e., $|x_{2,i}(t)| < c_{2,i}$, $t \geq 0$, is satisfied.

*Proof:* Consider the following Lyapunov function:

$$V_2 = V_1 + \frac{1}{2}\mathbf{z}_2^T\mathbf{z}_2 + \frac{1}{2}\tilde{\mathbf{d}}\mathbf{K}_d^{-1}\tilde{\mathbf{d}}. \tag{26}$$

Taking the derivative of $V_2$ yields that

$$\begin{aligned} \dot{V}_2 &= \dot{V}_1 + \mathbf{z}_2^T\dot{\mathbf{z}}_2 + \tilde{\mathbf{d}}\mathbf{K}_d^{-1}(\dot{\mathbf{d}} - \dot{\hat{\mathbf{d}}}) \\ &= -\mathbf{z}_1^T\mathbf{K}_1\mathbf{z}_1 - \mathbf{z}_1^T\boldsymbol{\Omega}_1\mathbf{G}_2\mathbf{z}_2 + \mathbf{z}_2^T\boldsymbol{\Omega}_2\left(\mathbf{M}^{-1}\boldsymbol{\tau} + \mathbf{f} + \mathbf{d}\right) \\ &\quad + \mathbf{z}_2^T\boldsymbol{\beta}_2 - \mathbf{z}_2^T\dot{\boldsymbol{\alpha}}_1 - \tilde{\mathbf{d}}\mathbf{K}_d^{-1}\dot{\hat{\mathbf{d}}} + \tilde{\mathbf{d}}\mathbf{K}_d^{-1}\dot{\mathbf{d}}. \end{aligned} \tag{27}$$

Substituting the controller (25) and the adaptive law (24) into (27), and according to *Lemma 2*, we have

$$
\begin{aligned}
\dot{V}_2 = & -\mathbf{z}_1^T \mathbf{K}_1 \mathbf{z}_1 - \mathbf{z}_2^T \mathbf{K}_2 \mathbf{z}_2 + \mathbf{z}_2^T \boldsymbol{\Omega}_2 \tilde{\mathbf{d}} \\
& - \tilde{\mathbf{d}}^T \mathbf{K}_d^{-1} \dot{\hat{\mathbf{d}}} + \tilde{\mathbf{d}}^T \mathbf{K}_d^{-1} \dot{\mathbf{d}} \\
= & -\mathbf{z}_1^T \mathbf{K}_1 \mathbf{z}_1 - \mathbf{z}_2^T \mathbf{K}_2 \mathbf{z}_2 + \mathbf{z}_2^T \boldsymbol{\Omega}_2 \tilde{\mathbf{d}} \\
& - \tilde{\mathbf{d}}^T \mathbf{K}_d^{-1} \left( \mathbf{K}_d \boldsymbol{\Omega}_2 \mathbf{z}_2 - \mathbf{B}_d \hat{\mathbf{d}} \right) + \tilde{\mathbf{d}}^T \mathbf{K}_d^{-1} \dot{\mathbf{d}} \\
\leq & -\mathbf{z}_1^T \mathbf{K}_1 \mathbf{z}_1 - \mathbf{z}_2^T \mathbf{K}_2 \mathbf{z}_2 - \frac{1}{2} \tilde{\mathbf{d}}^T \mathbf{K}_d^{-1} \left( \mathbf{B}_d - \mathbf{I}_n \right) \tilde{\mathbf{d}} \\
& + \frac{1}{2} \mathbf{d}^T \mathbf{K}_d^{-1} \mathbf{B}_d \mathbf{d} + \frac{1}{2} \dot{\mathbf{d}}^T \mathbf{K}_d^{-1} \dot{\mathbf{d}} \\
\leq & -\varphi V_2 + \eta
\end{aligned}
\tag{28}
$$

where

$$
\begin{aligned}
\varphi &= \min \left\{ 2\lambda_{\min}\left(\mathbf{K}_1\right), 2\lambda_{\min}\left(\mathbf{K}_2\right), \lambda_{\min}\left(\mathbf{B}_d - \mathbf{I}_n\right) \right\}, \\
\eta &= \frac{1}{2} \mathbf{d}^T \mathbf{K}_d^{-1} \mathbf{B}_d \mathbf{d} + \frac{1}{2} \dot{\mathbf{d}}^T \mathbf{K}_d^{-1} \dot{\mathbf{d}}.
\end{aligned}
\tag{29}
$$

According to *Lemma 1*, $V_2$ is UUB, and $V_2 \leq \eta/\varphi$ as $t \to \infty$. Now the following conclusions can be drawn:

1) $\mathbf{z}_1$, $\mathbf{z}_2$, and $\tilde{\mathbf{d}}$ are all UUB, which indicates that $\psi_{1,i} \in L_\infty$ and $\psi_{2,i} \in L_\infty (i = 1, \cdots, n)$.
2) Based on *Lemma 3*, for any initial joint position $x_{1,i}(0)$, $-\lambda_i(t) < e_i(t) < \lambda_i(t)$ holds for $t \in [T_p, +\infty)$.
3) Also from *Lemma 3*, if $|x_{2,i}(0)| < c_{2,i}$, the constraint on the velocity $-c_{2,i} < x_{2,i}(t) < c_{2,i}$ holds for $t \in [0, +\infty)$. ∎

*Remark 3:* It is worth pointing out that, if $h_{2,i} = 1$, the constraint on $x_{2,i}$ is the strongest, but it is also prone to singularity problem. However, if $h_{2,i} > 1$, although the constraint on $x_{2,i}$ is weakened, the risk of singularity problem can be reduced.

## IV. SIMULATION RESULTS

We conduct simulations based on the Quanser's QArm manipulator model to verify the proposed method. For details of the model, please refer to [20]. The QArm consists of 4 joints in a roll-pitch-pitch-roll configuration: the base (Q1), the shoulder (Q2), the elbow (Q3), and the wrist (Q4) joints, which is shown in Fig. 2. We only consider the motion of Q2 and Q3 in the simulation, i.e., the desired angular trajectories for Q1 and Q4 are $x_{d,1} = x_{d,4} = 0$ rad, for Q2 and Q3 are $x_{d,2} = 0.5\sin(4\pi t/15)$ rad and $x_{d,3} = 0.5\sin(2\pi t/15)$ rad. The model uncertainty is set to $\Delta \mathbf{M} = 0.2\mathbf{M}$, $\Delta \mathbf{C} = 0.1\mathbf{C}$, and $\Delta \mathbf{G} = 0.3\mathbf{G}$. The performance function for each joint is selected as $\gamma_i = 2$, $\lambda_{i,0} = 0.2$ rad, and $\lambda_{i,\infty} = 0.005$ rad $(i = 1, \cdots, 4)$. The parameters of the proposed controller are design as $T_p = 1$ s, $h_{u,i} = 3$, $h_{2,i} = 1.5$, $\mathbf{K}_1 = \text{diag}\{1, 11, 10, 1\}$, $\mathbf{K}_2 = \text{diag}\{1, 19, 21, 1\}$, $\mathbf{K}_d = \text{diag}\{1, 2000, 5000, 1\}$, and $\mathbf{B}_d = 2\mathbf{I}_4$. The simulations are run in the MatlabR2022b/Simulink environment with a 2 ms step size.

The angle tracking trajectories and the corresponding errors of Q2 and Q3 with different initial positions are shown in Fig. 3, where S1: $x_{1,2}(0) = x_{1,3}(0) = 0.3$ rad; S2: $x_{1,2}(0) = x_{1,3}(0) = 0.1$ rad; S3: $x_{1,2}(0) = x_{1,3}(0) = -0.3$

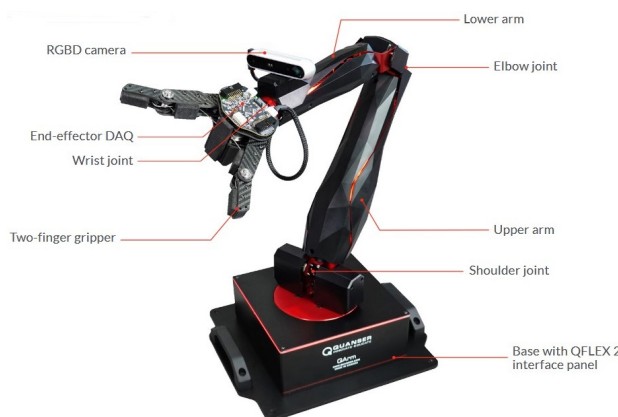

Fig. 2. Structure details of the QArm.

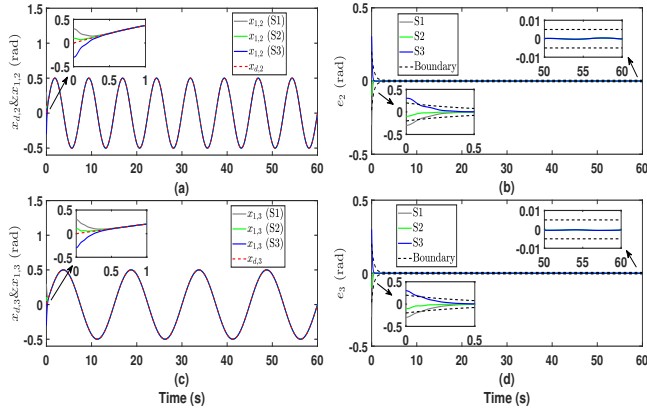

Fig. 3. Tracking performance with different initial joint positions in the simulation. (a) Tracking trajectory of Q2. (b) Tracking error of Q2. (c) Tracking trajectory of Q3. (d) Tracking error of Q3.

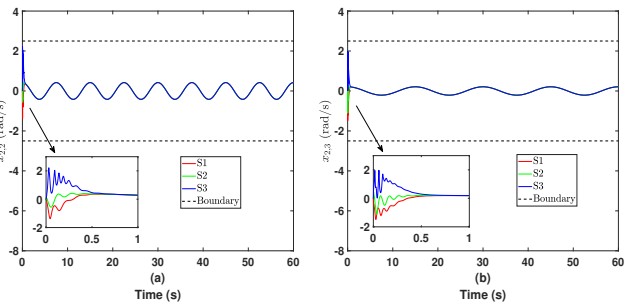

Fig. 4. Effects of velocity constraints in the simulation. (a) Joint Q2 velocity $x_{2,2}$. (b) Joint Q3 velocity $x_{2,3}$.

rad. The velocity constraints are $c_{2,2} = c_{2,3} = 2.5$ rad/s. We can see that the tracking errors of both joints converge quickly to the prescribed performance within the preset time $T_p$, regardless of the initial conditions of the manipulator. For situation S2, the tracking errors $e_2(t)$ and $e_3(t)$ remain within the corresponding performance boundaries $\pm\lambda_2(t)$ and $\pm\lambda_3(t)$ throughout the entire motion process. Furthermore, the joint velocities of Q2 and Q3 are shown in Fig. 4. It can be

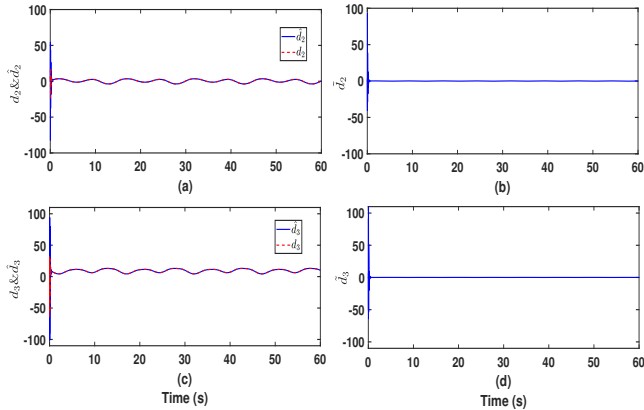

Fig. 5. Estimation effect of the lumped uncertainty. (a) Uncertainty $d_2$ and its estimate $\hat{d}_2$. (b) The estimation error $\tilde{d}_2$. (c) Uncertainty $d_3$ and its estimate $\hat{d}_3$. (d) The estimation error $\tilde{d}_3$.

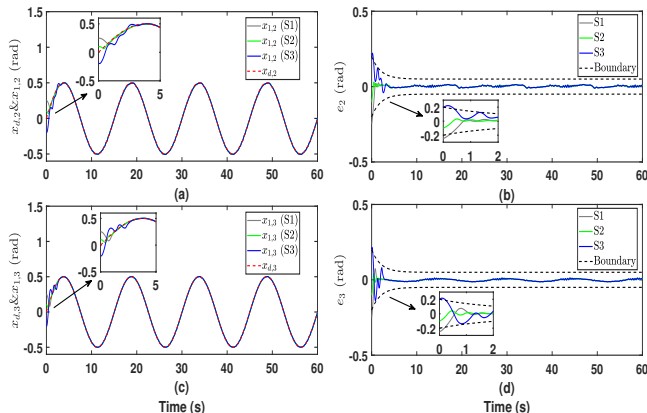

Fig. 6. Tracking performance with different initial joint positions in the experiment. (a) Tracking trajectory of Q2. (b) Tracking error of Q2. (c) Tracking trajectory of Q3. (d) Tracking error of Q3.

seen that $x_{2,2}(t)$ and $x_{2,3}(t)$ with any initial conditions are always within their constraint boundaries at any time, the peak value of $x_{2,2}(t)$ is 2.21 rad/s, and the peak value of $x_{2,3}(t)$ is 1.97 rad/s, which demonstrates the powerful state constraint capability of the proposed method. Finally, the estimation effect of the lupmed uncertainty is shown in Fig. 5, the initial joint positions are $x_{1,2}(0) = x_{1,3}(0) = 0.3$ rad. It is obvious that the designed adaptive law can estimate the uncertainties $d_2$ and $d_3$ of the two joints Q2 and Q3 very accurately, with an average estimation error of 0.0755 for $d_2$ and 0.0738 for $d_3$ in steady state (1 s- 60 s), which is of great help for the high-precision control of the manipulator.

## V. EXPERIMENTAL RESULTS

To further verify the effectiveness of the proposed method, we conduct relevant experiments on the QArm manipulator. It can be configured with the QFLEX 2 USB interface panel, which allows control and access from a computer via a USB connection. The control algorithm is developed in the Mat-labR2022b/Simulink environment equipped with the QUARC library. We also only consider the motion of Q2 and Q3 in the experiment, for each joint, the performance function is selected as $\gamma_i = 0.5$, $\lambda_{i,0} = 0.2$ rad, and $\lambda_{i,\infty} = 0.05$ rad. The controller parameters are designed as $T_p = 1$ s, $h_{u,i} = 3$, $h_{2,i} = 1.5$, $\mathbf{K}_1 = \text{diag}\{1, 25, 70, 1\}$, $\mathbf{K}_2 = \text{diag}\{1, 30, 55, 1\}$, $\mathbf{K}_d = \text{diag}\{1, 15, 15, 1\}$, and $\mathbf{B}_d = 2\mathbf{I}_4$.

The angle tracking trajectories and the corresponding errors of two joints with different initial positions are shown in Fig. 6, where S1: $x_{1,2}(0) = x_{1,3}(0) = 0.25$ rad; S2: $x_{1,2}(0) = x_{1,3}(0) = 0.1$ rad; S3: $x_{1,2}(0) = x_{1,3}(0) = -0.2$ rad. The desired trajectories are $x_{d,2} = x_{d,3} = 0.5\sin(2\pi t/15)$ rad, and the velocity constraints are set to $c_{2,2} = c_{2,3} = 1$ rad/s. For any initial conditions, we can see that the tracking errors of Q2 and Q3 converge to the prescribed performance within 1 s. Moreover, the velocities of both joints are also kept within the constraint boundaries, as shown in Fig. 7. Finally, the experimental scene of QArm motion process with signals display is shown in Fig. 8, where the desired trajectories are replaced

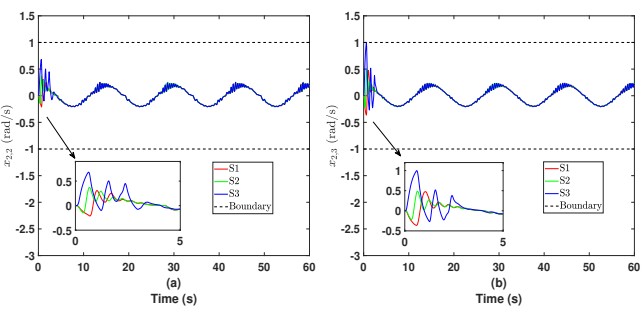

Fig. 7. Effects of velocity constraints in the experiment. (a) Joint Q2 velocity $x_{2,2}$. (b) Joint Q3 velocity $x_{2,3}$.

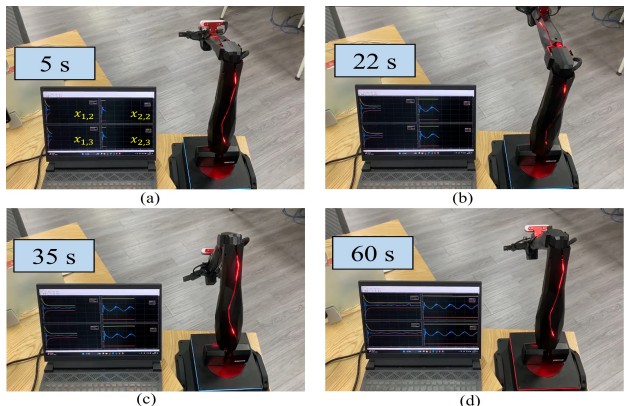

Fig. 8. Experimental scene of QArm motion process.

with $x_{d,2} = x_{d,3} = 0.5\sin(2\pi t/15) - 0.25(1 - t/30)$ rad, and the initial joint positions are $x_{1,2}(0) = x_{1,3}(0) = 0$ rad. From the laptop screen, the robotic manipulator remains stable during operation, and the steady-state tracking errors and joint velocities are always within the corresponding boundaries.

## VI. Conclusion

This paper presents an adaptive prescribed performance control method for the $n$-DOF manipulators with lumped uncertainty. We first design a normalization constraining function $g$ to transform the states and construct a new system model. The relationship between the constraint ability on tracking error or velocity and the risk of singularity problems can be balanced by adjusting the parameters of $g$. Then develop a prescribed performance controller with an adaptive law, which can guarantee that the tracking error reaches the ideal accuracy from any initial joint positions, and the lumped uncertainty in the system is estimated and compensated by the adaptive law. Moreover, the velocity constraint can be independently imposed on any joint. Finally, the effectiveness and superior control performance of the proposed method are verified through simulations and experiments on the QArm robotic manipulator.

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
