# OpenReview forum: "Adaptive Prescribed Performance Control of Robotic Manipulators with Velocity Constraints and Arbitrary Initial Joint Positions"
_IEEE.org/ICIST/2024/Conference — IEEE ICIST 2024 Conference Submission_

### Official Review · Reviewer_kDsM · 2024-08-21
**The manuscript is well-structured and provides valuable insights into prescribed performance control of robotic manipulators. The research is thorough and methodologically sound, with clear and relevant results. I recommend the paper for publication.**

**Rating:** 7
**Confidence:** 5

**Review:**

1.The research is conducted with rigor, and the analysis is both comprehensive and methodologically sound. The results are well-presented and supported by the data, demonstrating a clear understanding of the subject matter.
2.The findings of the study are highly relevant to the field and offer practical implications for prescribed performance control. The potential impact of the work on current practices or future research directions is considerable

---

### Official Review · Reviewer_Y4oK · 2024-08-24
**reasonable motivation，reasonable innovation**

**Rating:** 8
**Confidence:** 4

**Review:**

In the manuscript titled "Adaptive prescribed performance control of robotic manipulators with velocity constraints and arbitrary initial joint positions",Xing Ren et al.performed novel adaptive prescribed performance control method for the n-DOF robotic manipulators and demonstrated normalization constraining function can reduce the risk of singularity problems ,and the developed adaptive prescribed performance controller and adaptive law can guarantee that the tracking error reaches the desired accuracy within the preset time and improves the control performance. This study contains some interesting findings and are valuable for the understanding of reducing the risk of singularity problems,and sufficient literature comparison leads to the motivation of this paper. Maybe the current manuscript can be polished by a native English speaker or a professional language editing service.

---

### Official Review · Reviewer_3Ce1 · 2024-08-25
**The research content of this paper is to propose a new adaptive predetermined performance control method for n-DOF robots. By adjusting the parameters of the function g, you can reduce the risk of singular problems. And the designed controller allows the initial joint position of the manipulator to be arbitrary, and can ensure that the tracking error reaches the expected accuracy within the preset time TP. In addition, different velocity constraints are applied to each joint individually, and adaptive laws are designed to estimate the concentration uncertainty in the system and compensate for its negative effects, which improves the control performance. Sufficient theoretical explanations and simulation verifications are carried out. The advantage is that the control performance is really improved and the preset time performance is guaranteed. However.1. Please explain what assumptions were used to justify them. 2. Whether the future work will be explained.**

**Rating:** 8
**Confidence:** 4

**Review:**

The research content of this paper is to propose a new adaptive predetermined performance control method for n-DOF robots. Firstly, a normalized constraint function g is designed for state transition and the construction of a new system model.By adjusting the parameters of the function g, you can reduce the risk of singular problems. Then, based on the constructed system model, an adaptive performance controller was developed, which allowed the initial joint position of the manipulator to be arbitrary, and could ensure that the tracking error reached the desired accuracy within the preset time TP.In addition, different velocity constraints are applied to each joint individually, and adaptive laws are designed to estimate the concentration uncertainty in the system and compensate for its negative effects, which improves the control performance.Finally, the effectiveness of the proposed method is verified by simulations and experiments of a four-degree-of-freedom manipulator.
The advantage is that it does improve the control performance.
However.1. Please explain what assumptions were used to justify them.
2. Whether the future work will be explained.

---

### Decision · Program_Chairs · 2024-09-08

Accept (Oral)